# Laser-induced graphitization of polydopamine leads to enhanced mechanical performance while preserving multifunctionality

Kyueui Lee [1,8], Minok Park[2,8], Katerina G. Malollari[3], Jisoo Shin[1], Sally M. Winkler [1], Yuting Zheng[4], Jung Hwan Park[5], Costas P. Grigoropoulos [2✉] & Phillip B. Messersmith [1,6,7✉]

Polydopamine (PDA) is a simple and versatile conformal coating material that has been proposed for a variety of uses; however in practice its performance is often hindered by poor mechanical properties and high roughness. Here, we show that blue-diode laser annealing dramatically improves mechanical performance and reduces roughness of PDA coatings. Laser-annealed PDA (LAPDA) was shown to be >100-fold more scratch resistant than pristine PDA and even better than hard inorganic substrates, which we attribute to partial graphitization and covalent coupling between PDA subunits during annealing. Moreover, laser annealing provides these benefits while preserving other attractive properties of PDA, as demonstrated by the superior biofouling resistance of antifouling polymer-grafted LAPDA compared to PDA modified with the same polymer. Our work suggests that laser annealing may allow the use of PDA in mechanically demanding applications previously considered inaccessible, without sacrificing the functional versatility that is so characteristic of PDA.

[1] Department of Bioengineering, University of California at Berkeley, Berkeley, CA 94720, USA. [2] Laser Thermal Laboratory, Department of Mechanical Engineering, University of California at Berkeley, Berkeley, CA 94720, USA. [3] Department of Mechanical Engineering, University of California at Berkeley, Berkeley, CA 94720, USA. [4] Department of Chemical Engineering, University of California at Berkeley, Berkeley, CA 94720, USA. [5] Department of Mechanical Design Engineering, Kumoh National Institute of Technology, 61 Daehak-ro, Gumi, Gyeongbuk 39177, Republic of Korea. [6] Department of Materials Science and Engineering, University of California at Berkeley, Berkeley, CA 94720, USA. [7] Materials Sciences Division, Lawrence Berkeley National Laboratory, Berkeley, CA 94720, USA. [8] These authors contributed equally: Kyueui Lee, Minok Park. ✉email: cgrigoro@berkeley.edu; philm@berkeley.edu

Functional surface coatings are often applied onto modern engineered materials to achieve augmented and altered performance. Of particular interest is polydopamine (PDA), a coating material that has many appealing characteristics such as the ability to coat most inorganic and organic materials, low toxicity, one-step deposition using a simple method, and multiple functional properties[1–3]. Hence, since its inception in 2007, its utilization has grown extensively in numerous fields of engineering, including biomaterials[4,5], energy storage and harvesting devices[6], photonics[7,8], and medical therapeutics[9].

Nevertheless, practical applications and commercialization of PDA have been substantially limited by two main problems. The first is inherently poor mechanical properties of PDA, which inevitably results in continuous abrasion of PDA film, exposing the underlying substrate in the end[2]. Considering the current understanding of the formation and molecular structure of PDA[10–12], this deficiency could be attributed to the relatively weak noncovalent intermolecular bonds between PDA building blocks[13–15]. Another significant problem is the unavoidable spontaneous deposition of PDA particles ranging from hundreds of nanometers to a few microns on top of the PDA layer, producing high roughness that may be undesirable from a surface engineering perspective.

Many prior efforts have been made to address the aforementioned challenges. For example, posttreatment approaches have been proposed to transform mechanical characteristics of the PDA, including mild thermal annealing[16], molecular integration of metal ions[17], and introduction of a cross-linker[18]. However, even these methods do not offer dramatic improvements in wear resistance. With respect to the roughness problem, synthesis of PDA films has been attempted by accelerating the kinetics of oxidative polymerization[19] or by using alternative building blocks such as norepinephrine[20]. However, undesirable particle formation is inevitable even with such approaches.

Photon-mediated heating, within the concentrated area and time rendered by a laser, is an important tool to precisely control as well as improve the physicochemical properties of target materials via molecular reconstruction and phase transformation[21]. Laser processing can access regimes that are not achievable via conventional thermal treatment. In particular, blue-diode laser[22] has been introduced in thin film annealing owing to its scalability, continuous wave characteristics, high optical power density per unit area or volume, low cost, and high photon energy (2.82 eV)[23,24]. Considering that the light absorption of PDA is in the range of blue-diode laser wavelength (440 nm)[15], we inferred that blue-diode laser annealing (BLA) could be employed to efficiently anneal and improve PDA properties.

In this work, we demonstrated that laser-annealed PDA (LAPDA) shows exceptional enhancement in both wear resistance and roughness compared to pristine PDA. Specifically, the BLA process photo-thermally induces partial graphitization of the PDA film through a transient heating process inducing peak temperatures over 1000 K. Hence, the partially graphitized PDA film showed substantially increased wear resistance through the replacement of noncovalent intermolecular bonds with covalent ones, while still retaining the catechol functionality that is inherent to PDA. Moreover, the level of graphitization can be controlled simply by adjusting the laser power to achieve a balance between the wear resistance and the catechol functionality of the PDA film. In contrast, bulk thermal annealing of PDA in a furnace typically results in fully graphitized film with poor resemblance to PDA[25]. Scratch resistance tests with diamond tips confirmed that the scratch resistance of LAPDA can even surpass the wear resistance of hard inorganic materials ($TiO_2$ and $SiO_2$). Furthermore, the BLA process selectively eliminates randomly agglomerated PDA nanoparticles (NPs) from the surface, resulting in dramatic reduction of surface roughness. Lastly, the ability of LAPDA films to host secondary surface functionalization was demonstrated by grafting polyethylene glycol (PEG) onto the surface. Unlike PEGylated PDA films that were prone to scratch defects, the PEGylated LAPDA preserved antifouling characteristics over an entire wafer-sized area (1″ by 1″).

## Results

Shown in Fig. 1a, Supplementary Fig. 1, and Supplementary Movie 1 are schematics of the continuous wave BLA process for the PDA film in the ambient air condition. The PDA film with 248 nm average thickness was prepared by dip-coating quartz substrate in the aqueous dopamine hydrochloride solution for 24 h (see "Methods" section). The focused blue-diode laser has an elliptically shaped beam profile with 99 μm long axis and 14 μm short axis based on $e^2$ beam diameter, measured by a knife edge experiment. Annealing of a PDA film deposited on a 1″ by 1″ quartz substrate was accomplished by raster-scanning with a speed of 50 mm/s (equivalent to dwell time of 280 μs $= \frac{short\ axis}{scanning\ speed}$). The scan lateral separation was set at 35 μm in order to achieve uniform annealing on the overlapped region, as illustrated in Supplementary Fig. 1b.

To corroborate the partial graphitization by the BLA process, the Raman spectra of the pristine PDA and the LAPDA films with various laser annealing powers (0.9, 1.2, 1.9, 2.4, and 2.9 W) were compared (Fig. 1b), and the corresponding laser intensity (kW/$cm^2$) is presented in Supplementary Table 1. The Raman spectrum of the pristine PDA shows broad D (disordered carbon, around 1365 $cm^{-1}$) and G (graphitic carbon, near 1575 $cm^{-1}$) peaks consonant with the film's amorphous state. However, these peaks become more distinct in the LAPDA films as the laser power increases. The graphitization of carbon-based materials is reflected by the change in the $I_D/I_G$ ratio and the Raman shift of G peak[26–28]. As the power of the blue-diode laser increases, the $I_D/I_G$ ratio of LAPDA films increases from ~0.5 to ~1.0, and the G peaks shift from ~1579 to ~1605 $cm^{-1}$ (Fig. 1c). These trends indicate that the amorphous PDA film is gradually converted into the graphitic material consisting of nanocrystallites[26–28]. To investigate changes in the atomic composition of the films, X-ray photoelectron spectroscopy (XPS) analysis of the PDA and the LAPDA films was performed. Figure 1d shows the high-resolution C1s data collected from the pristine PDA and the LAPDA films, achieved by different laser annealing powers. After the BLA process, there was a relative increase in the peaks from the carbon–carbon bonds ($sp^2$ and $sp^3$) compared to the peaks from nitrogenated and oxygenated groups. To quantify the detailed composition of the carbon–carbon bonds, the C1s peaks were deconvoluted by the representative bonds in the PDA film: (i) $sp^2$, $sp^3$ bonds at ~284.5 eV, (ii) C–O, C–N bonds at ~286 eV, (iii) C=O bond at ~287.5 eV, and (iv) π bond at ~289 eV (Supplementary Fig. 2)[12]. Consequently, the atomic percentage (at.%) of carbon–carbon bonds ($sp^2$, $sp^3$) increases from 47 to 74 at.%, which is in proportion to the laser power (Fig. 1e and Supplementary Table 2). On the other hand, the composition of nitrogenated and oxygenated functional groups decreases from 45 at.% (in the pristine PDA) to 19 at.% (in the LAPDA at 2.9 W). This decreasing trend is also confirmed in the XPS survey analysis of the PDA and LAPDA films (Supplementary Fig. 3 and Supplementary Table 3). Considering the Raman and XPS results, it is verified that the nitrogenated and oxygenated atomic defects in PDA are partially eliminated by the BLA process and the degree of the graphitization can be controlled by adjusting the laser power.

Previously, it was found that the graphitization of the PDA gives rise to molecular reorganization through additional covalent

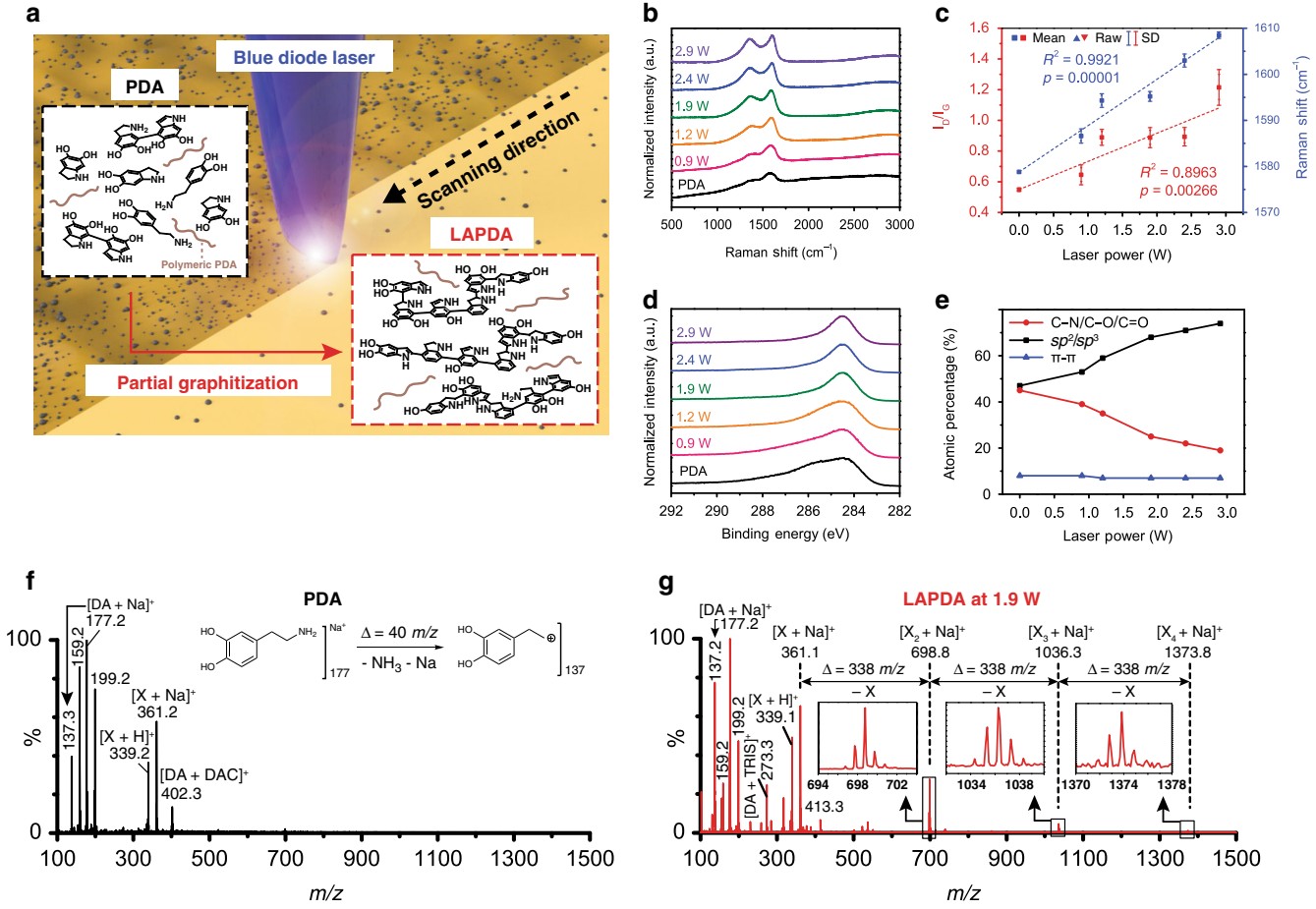

**Fig. 1 Partial graphitization of PDA film by BLA process for mechanical enhancement. a** Schematic of the BLA process in which pristine PDA film is raster-scanned by laser light. Graphitization level of each LAPDA film was characterized by **b, c** Raman and **d, e** X-ray photoelectron spectroscopies. Rectangular, triangular marks, and error bars in Fig. 2c represent mean values, original data points, and standard deviation, respectively. MALDI-TOF molecular mass of **f** PDA film and **g** LAPDA film at 1.9 W. The delta (uppercase) means a mass change. Source data are provided as a Source Data file.

conjugations between PDA building blocks, culminating eventually in increased crystallinity[26,29]. To closely examine this in PDA and LAPDA films, matrix-assisted laser desorption/ionization time of flight (MALDI-TOF) mass spectrometry was employed. In the pristine PDA film, the peaks from dopamine derivatives were identified: an unreacted dopamine-sodium complex (177.2 $m/z$), its fragment (137.3 $m/z$)[30], and a dopamine/dopaminochrome complex derivative (402 $m/z$)[31] (Fig. 1f). Furthermore, a protonated unknown compound ($[X + H]^+ = 339 \, m/z$) and its sodium complex ($[X + Na]^+ = 361 \, m/z$) were observed. Although the chemical structure of the unknown compound X is not clearly assigned herein, it is likely to be a derivative of dimerized 5,6-dihydroxyindole which is generally accepted as a major building block contributing to PDA assembly.[10,11,32] In the LAPDA at 1.9 W, the additional peaks from the high molecular weight region are observed at 699, 1036, and 1374 $m/z$, and obviously, there is a repetitive mass change ($\Delta = 338 \, m/z$) matched with the fragmented X between those peaks (Fig. 1g). This fragmentation pattern is analogous to mass spectrometry of polysaccharides where the mass change corresponds to the basic building block (i.e., monosaccharides)[33]. Thus, the molecules in the higher molecular weight regime are considered to be tetramer (699 $m/z$), hexamer (1036 $m/z$), and octamer (1374 $m/z$) of the 5,6-dihydroxyindole derivative, which strongly supports that the BLA process activates the further conjugation between the PDA building blocks, as illustrated schematically in Fig. 1a. To verify the suggested model, UV–VIS spectroscopy was

performed. The red-shifted UV–VIS absorption spectrum of the LAPDA at 1.9 W compared to the pristine PDA indicates an elongated π-conjugation system[34], supporting the formation of additional covalent bonds in the molecular structures of the PDA by the BLA process (Supplementary Fig. 4).

To estimate the temperature field reached during laser annealing, the 3D heat conduction equation was numerically solved (Supplementary Fig. 5a). Supplementary Fig. 5b shows the time-dependent temperature profiles and Supplementary Fig. 5c indicates the temperature field contours at the surface of the PDA film. Detailed analysis regarding the photothermal process is discussed in Supplementary Fig. 6. The maximum transient temperature reached in the PDA film is predicted to be 853, 1000, 1336, 1584, and 1836 K, under 0.9, 1.2, 1.9, 2.4, and 2.9 W, respectively. Whereas annealing near ~853 K (at 0.9 W) does not yield a noticeable change in graphitization level of the PDA film, heating to over 1000 K (achieved from 1.2 to 2.9 W) leads to graphitization. Thus, achieving elevated temperatures for hundreds of microseconds duration is sufficient for graphitization of the PDA film, in contrast with the thermal annealing over 1000 K in the furnace[25,29].

To assess the effect of the partial graphitization on the wear resistance of PDA, scratch resistance tests on pristine and LAPDA samples were performed. During testing, a conospherical indenter was subjected to a constant or progressive load while moving across the sample surface[35]. Figure 2a and Supplementary Fig. 7a summarize the scratching experiments using 50 μN normal

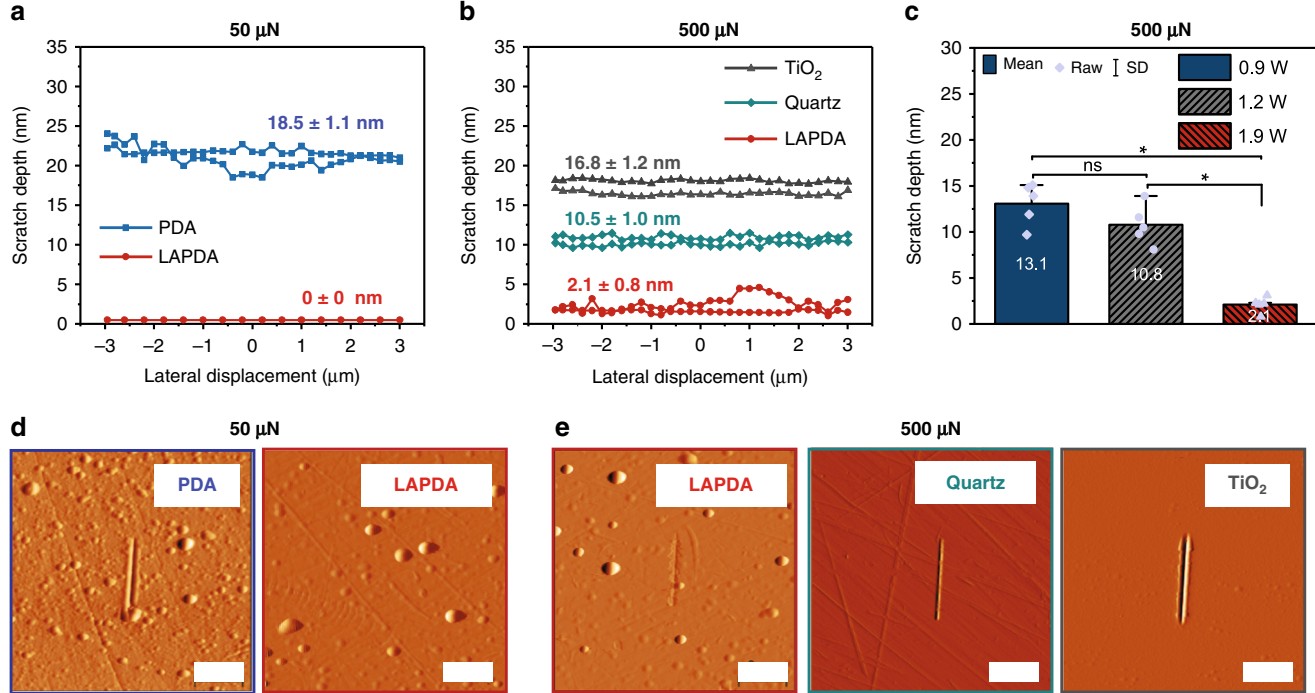

**Fig. 2 Scratch resistance measurements on pristine PDA film, LAPDA film, and inorganic substrates.** Representative scratch depth–lateral displacement curves at **a** 50 μN and **b** 500 μN, respectively. **c** Scratch depths for LAPDA films annealed at 0.9, 1.2, and 1.9 W were compared. Bars represent mean values and original data points overlaid as purple dots; error bars represent standard deviation. Statistical significance (asterisk symbol) was assessed using a one-way analysis of variance with Tukey's post-test (n = 5). **d**, **e** Corresponding scratched images on the substrates of (**a**, **b**). The scale bar is 2 μm. Source data are provided as a Source Data file.

load performed on the pristine PDA film and the LAPDA film annealed at 1.9 W. The average scratch depth for the pristine PDA at 50 μN normal load was 18.5 ± 1.1 nm, whereas the LAPDA films were undamaged at this load. Scratches in LAPDA films were only observed at a much higher load as shown in Fig. 2b and Supplementary Fig. 7b. At 500 μN, the scratch depth of a 37 nm thick LAPDA film was only 2.1 ± 0.8 nm, whereas the scratch depth of a pristine PDA film under the same conditions was the full thickness of the film (248 nm). This represents more than a 100-fold increase in scratch resistance, which is more effective than the conventional thermal annealing approach showing a 2.5-fold increase[16]. To further put the scratch resistance of LAPDA in perspective, we also performed scratching experiments on bare TiO$_2$ and quartz substrates at a load of 500 μN, producing average scratch depths of 16.8 ± 1.2 nm for TiO$_2$ and 10.5 ± 1.0 nm for quartz. Surprisingly, this indicates that the exceptional scratch resistance of the organic LAPDA film is several-fold better than even hard inorganic solids. Moreover, the estimated hardness of LAPDA (6.6 ± 1.0 GPa) was superior to that of TiO$_2$ and quartz, showing 4.6 ± 0.4 and 6.2 ± 0.9 GPa, respectively (Supplementary Fig. 8). The influence of laser power on scratch resistance of the LAPDA films is shown in Fig. 2c. Increasing laser power leads to decreased scratch depth (13.1 nm at 0.9 W, 10.8 nm at 1.2 W, and 2.1 nm at 1.9 W, respectively), which shows that the scratch resistance can be manipulated by simply adjusting the laser power.

As discussed above, Raman, XPS, and MALDI measurements indicate graphitization and increase in crystallinity of the PDA, which could explain the enhanced scratch resistance resulting from laser annealing. In addition, it is well documented that ~20% of PDA is composed of unpolymerized monomeric and/or partially polymerized oligomeric species[13], and it has been shown that thermal annealing can induce subsequent polymerization of these species[16]. As shown in Fig. 1f, g, a similar trend was

observed for LAPDA where higher molecular weight species are detected. We speculate that further polymerization of PDA during laser annealing produces additional strong chemical bonds between subunits and promotes mechanical interlocking of the structure, leading to considerable enhancement of the mechanical performance of the coatings.

To examine the change in the surface morphology, each sample was characterized by atomic force microscopy (AFM), optical microscopy, and scanning electron microscopy (SEM), as shown in Fig. 3. The pristine PDA film and the LAPDA film annealed at 1.9 W were compared in Fig. 3a, b, and all measurements of each sample are summarized in Fig. 3c. The pristine PDA film shows an average thickness of 248 nm with high roughness (~230 nm) due to the agglomerated NPs (Fig. 3a). In contrast, after the laser annealing under 1.9 W, most NPs on the surface were ablated, resulting in exceptionally smoothened surface with 16 nm roughness and decreased thickness with 37 nm (Fig. 3b), which is also identified in macroscopic and microscopic scales (Fig. 3d and Supplementary Fig. 9). We expect that the initial thickness of the PDA film would not have a huge impact on the improvement in the smoothness since the main contributor for roughness is the randomly agglomerated NPs.

The mechanism of film smoothing during laser annealing appears to involve selective NP ablation from the surface due to two main reasons. (i) Based on Mie theory prediction displayed in Supplementary Fig. 10[36], spherically shaped PDA NPs have substantially better absorption efficiencies (0.25–0.95) than the absorptance (0.05) of the flat PDA film. This indicates that PDA NPs theoretically absorb more incident laser energy than the flat PDA film. (ii) Since the PDA NPs have limited interfacial contact and correspondingly large thermal contact resistance, they can be preferentially heated and ablated. To verify the annealing effect on PDA NPs, a high-speed video with 8000 fps was recorded for the ablation process at 1.9 W (Supplementary

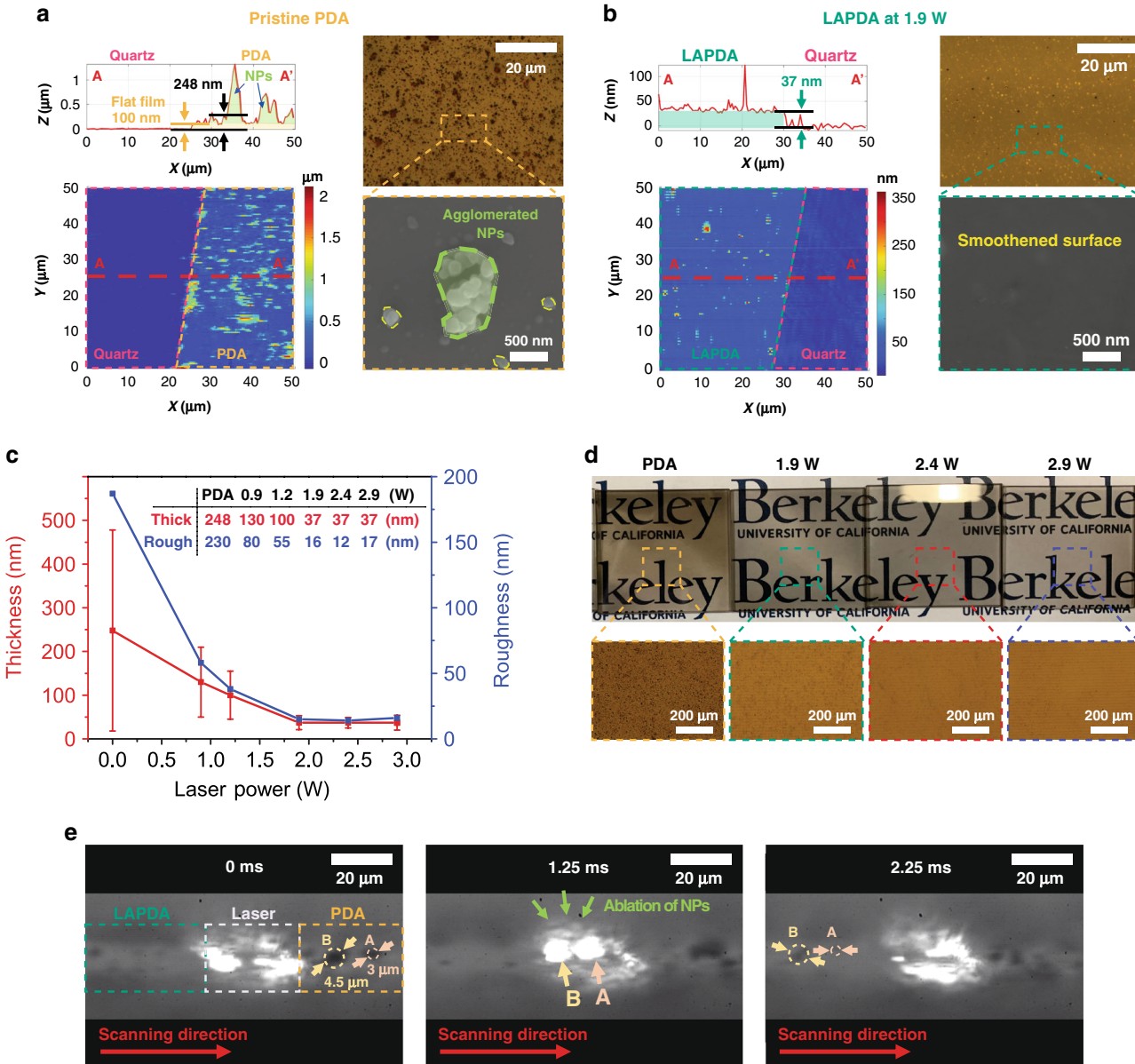

**Fig. 3 Reduced roughness in LAPDA films due to thermal ablation of PDA NPs.** AFM images with line-scan profiles, optical microscope images, and SEM images of **a** pristine PDA and **b** LAPDA at 1.9 W. **c** Thickness/roughness profiles and **d** corresponding digital and optical images of LAPDA films annealed at different laser powers: 1.9, 2.4, and 2.9 W. Error bars represent standard deviation. **e** Time-resolved snapshots of annealing process under 1.9 W recorded by a high-speed camera. Source data are provided as a Source Data file.

Movie 2). From the time-resolved snapshots taken from the video, it is found that 3 μm agglomerated NPs (marked as A) were completely eliminated, and ~4.5 μm NPs (marked as B) were almost ablated after the BLA (Fig. 3e). Through a lumped capacitance analysis (see Supplementary Fig. 11 for the full derivation), the 3 μm NPs could be heated to ~2900 K, and the ~4.5 μm NPs could be annealed to ~2100 K. Consequently, the deposited laser energy is converted into heat and confined within the NP causing detachment of the NP either by differential thermal expansion or evaporation/fragmentation. In contrast, the absorbed heat in flat PDA film is mainly dissipated into the substrate (Supplementary Figs. 5 and 11b). Thus, in addition to the aforementioned chemical and physical transformations induced in the continuous PDA film the weakly bound PDA NPs are removed, leading to the smoothened surface in the LAPDA film.

Finally, it is important to show that enhancement of PDA mechanical performance achieved by laser annealing was not achieved at the expense of other important properties of PDA, such as the ability to undergo secondary surface functionalization. Accordingly, we assessed antifouling performance of the coatings after grafting of an antifouling polymer onto LAPDA through nucleophile–catechol conjugation via Michael type addition, an approach that has been widely employed in PDA post modification[37]. Thus, a representative nucleophilic polymer (methoxy PEG thiol) was successfully grafted on the LAPDA through covalent conjugation between catechols and thiols (see Supplementary Fig. 12 for the detailed methods). Moreover, an additional example of secondary surface modification via electroless metallization on LAPDA was presented in Supplementary Fig. 13.

To investigate the antifouling performance of PEGylated LAPDA, we quantitatively compared the bacterial attachment

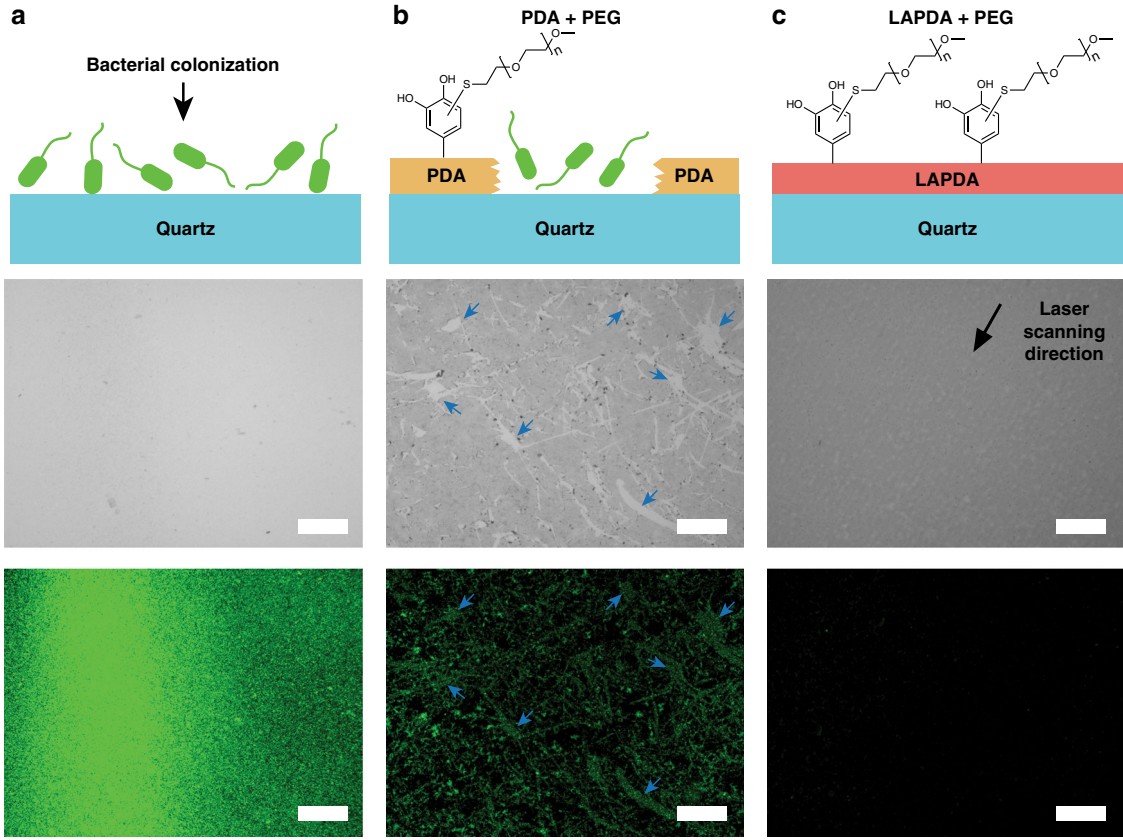

**Fig. 4 Preserved catechol functionality in LAPDA.** Antifouling effect on **a** bare quartz, **b** PEGylated PDA, and **c** PEGylated LAPDA. Schematics on top illustrate bacterial colonization behavior (*E. coli.*) on the surfaces. Middle and bottom images are corresponding bright-field and fluorescence images, respectively. The blue arrows indicate bacterial accumulation on sites of mechanical damage in the PEGylated PDA coating. The scale bar is 200 μm.

driven by *Escherichia coli* (*E. coli*) from the three different substrates: bare quartz, the PEGylated PDA film on quartz, and the PEGylated LAPDA film on quartz. For this, bacteria were seeded on the substrates and incubated for 24 h. Then, all the samples were washed with the saline to remove unattached bacteria and stained with bacterial LIVE/DEAD stain. As shown in Fig. 4a, the bare quartz substrate (positive control) was fully occupied by *E. coli* (green spots) within 24 h. In contrast to pristine PDA which performed poorly due to weak mechanical adhesion to the substrate (Fig. 4b and Supplementary Fig. 14), under the same conditions the PEGylated LAPDA film remained smooth, mechanically stable and showed successful inhibition of bacterial attachment (94% reduction relative to bare quartz) (Fig. 4c and Supplementary Fig. 15).

In summary, a mechanically strengthened and smoothened PDA film was achieved through the BLA process. Specifically, the laser-induced partial graphitization plays a pivotal role in the mechanically strong LAPDA film formation, and the graphitization level can be easily controlled by modulating the laser power. Scratch tests with diamond tips revealed that LAPDA films performed even better than inorganic substrates. Furthermore, the roughness of LAPDA film can be reduced from ~230 to ~15 nm as a result of selective NP ablation during the BLA process as shown by high-speed in situ recording of the PDA surface and lumped capacitance analysis. Finally, preservation of the attractive secondary modification properties of PDA was shown through polymer grafting and antifouling assays. From the processing point of view, the film annealing throughput can be increased since multiple blue-diode laser modules can be combined to form a wide beam line of high power. We believe that combining the exceptional

wear resistance of LAPDA films with the attractive secondary modification features of conventional PDA will open many new opportunities for PDA as an advanced surface engineering material.

## Methods

**Materials**. Quartz substrate was purchased from Ted Pella, Inc. (USA). Tris base was purchased from J.T.Baker (USA). *E. coli* (25922) was purchased from ATCC (USA). Corning cellgro phosphate-buffered saline was purchased from Mediatech (USA). Bacterial LIVE/DEAD staining kit was purchased from Invitrogen (USA). Dopamine hydrochloride, 2,5-dihydroxybenzoic acid and all other reagents were purchased from Sigma-Aldrich (USA).

**PDA deposition protocol**. A 1″ by 1″ quartz substrate was sonicated in isopropanol followed by deionized water for 10 min each, followed by evaporation under a stream of nitrogen gas. This clean quartz substrate was immersed for 24 h in a solution of 2 mg/mL of dopamine hydrochloride dissolved in 10 mM Tris base (pH 8.5 adjusted by adding 1 M NaOH) in a mild shaking condition (~50 rpm), resulting in ~100-nm-thick PDA film with randomly distributed PDA NPs on the substrate. The PDA coated substrate was rinsed with deionized water and dried with nitrogen gas before use.

**BLA of PDA**. A blue-diode laser was collimated by two convex lenses and irradiated through a ×5 objective lens (NA 0.14, Mitutoyo, Japan) on the pristine PDA film (Supplementary Fig. 1). The PDA film was placed on PC controllable high-precision XYZ positioning stages (ANT95-XY-MP, and ANT95-50-L-Z-RH, Aerotech, USA) to anneal the entire surface. To monitor annealing process in situ, the PDA film was illuminated by white light from one side, and a high-speed charge-coupled device camera (FASTCAM Mini UX50, Photron, Japan) equipped with a ×10 objective lens (NA 0.28, Mitutoyo, Japan) was installed in a grazing angle on the other side.

**Scratch testing**. Nanoscratch tests were performed using Hysitron TI-950 TriboIndenter (Bruker, USA) equipped with a 2-D transducer and a 300 nm conospherical probe. The scratch load function consisted of three distinct segments: a

trace segment to determine the surface profile at the site of the scratch, a 5 µm long scratch segment, and a retrace segment to determine the residual deformation after the load was removed. In this study, the residual depth is referred to as scratch depth. To perform scratch testing measurements, two scratch test load functions were used: one with a constant normal load of 50 µN and the other with a constant normal load of 500 µN. Multiple scratches ($n = 15$) for each load function were performed on three samples of pristine PDA and LAPDA coatings to determine variation between films. In addition, multiple tests ($n = 8$) for the 500 µN load function were also performed on the TiO$_2$ and quartz control samples. For all scratch tests, a tilt correction was performed on the scratch test data using the trace segment of the load function. Finally, using Hysitron's SPM imaging technique, the surface topography over an area of 15 µm by 15 µm was mapped at each test site.

**X-ray photoelectron spectroscopy**. XPS spectra of the pristine PDA and LAPDA films were obtained using Phi 5600 XPS (Perkin Elmer, USA) to analyze atomic compositions. A monochromatic X-ray source (Al Kα) was used for the data collection. To avoid the effect of surface charging, the neutralizer was applied simultaneously to the surfaces with a constant current (~1 µA). The at.%s were calculated by deconvolution of the XPS spectra with MATLAB-based software (MultiPak) provided by Physical Electronics, Inc.

**Matrix-assisted laser desorption/ionization time of-flight**. A matrix solution was made by referring to the previous method used for PDA analysis[30]. Specifically, 2,5-dihydroxybenzoic acid was dissolved at a concentration of 20 mg/mL in a solution of 0.1% trifluoroacetic acid and acetonitrile in a volume ratio of 1:4. The analytes (PDA and LAPDA films) deposited on the quartz substrate were exfoliated from the surface with stainless steel spatulas. Then, ~10 µL of matrix solution was added to the exfoliated samples. One microliter of the solution mixture was transferred to the MALDI plate and dried under ambient conditions. The MALDI-TOF mass spectrometer (Voyager-DE Pro, Applied Biosystems, USA) equipped with a 337 nm nitrogen laser was applied for the measurement. All the collected data were achieved from the reflector mode with the accelerating voltage of 20 kV.

**UV–VIS spectroscopy**. To measure the UV–VIS spectra of PDA and LAPDA films on quartz, each analyte was fixed to the sample cuvette holder area, and the bare quartz was simultaneously applied to the reference cuvette holder area. The UV–VIS spectra were taken by a UV2600 spectrophotometer (Shimadzu Scientific Instruments, Japan). The absorbance spectra were normalized using an Origin Pro software (Version 8, OriginLab, USA).

**Antifouling tests on PEGylated substrates**. The antifouling ability of the substrates was evaluated using *E. coli*. In detail, *E. coli* was cultured in Luria–Bertani broth for 16 h at 37 °C in a rotary shaker (Incu-Shaker Mini, Benchmark Scientific, USA), with 160 rpm. After the overnight culture, *E. coli* were centrifuged at 4000 rpm for 5 min and the pellets were resuspended in phosphate-buffered saline. For a bacterial adhesion testing, sample substrates were placed in each well of a 24-well tissue culture plate and immersed in a bacterial solution (1 mL of $1 \times 10^9$ cells/mL). After incubation under static conditions for 24 h at 37 °C, the samples were washed three times with saline and subsequently stained with a bacterial LIVE/DEAD staining kit (Invitrogen, USA) according to manufacturer's instructions. Stained *E. coli* were observed using a BZ-X800 fluorescence microscope (Keyence, USA). The entire surface of the sample was acquired, and the area covered by bacteria was quantified using Image J software (National Institutes of Health, USA). All the values were normalized to the negative control (bacterial attachment on bare quartz = 1.0).

**Statistical information**. The statistical significance in Fig. 2c was assessed using a one-way analysis of variance with Tukey's post-test using Origin software. The $p$ values indicated in Supplementary Fig. 15 were calculated by the unpaired two-tailed $t$-tests with equal variances using Excel software. $p < 0.05$ indicates statistical significance.

## Data availability
The authors declare that the data supporting the findings of this study are available within the article and its Supplementary Information files. Source data are provided with this paper.

## Code availability
The codes that support the findings of this study are available from the corresponding authors upon reasonable request.

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

## Acknowledgements
This work was supported by the National Institutes of Health Grant R37 DE014193. Work at the Laser Thermal Laboratory was supported by Laser Prismatics LLC.

## Author contributions
P.B.M. and C.P.G. conceived this study. K.L., M.P., K.G.M., J.S., S.M.W., Y.Z., and J.H.P. performed experiments, processed data, and wrote the manuscript. P.B.M. and C.P.G. contributed to the discussion and revised the manuscript. All authors have given approval to the final version of the manuscript.

## Competing interests
The authors declare no competing interests.
