## [Peer Review File · Nature Communications]

REVIEWER COMMENTS

Reviewer #1 (Remarks to the Author):

Lee and co-workers report a novel processing method for coating surfaces with an ultra scratch resistant and chemically functional layer of polydopamine. The technique relies on diode laser induced annealing of a pre-coated layer of polydopamine (PDA). Superior scratch resistance is a result of graphitization and smoothening of the PDA layer. The simplicity of adjusting laser power to control graphitization and hence-wear resistance is very beneficial. The fact that this is an ambient, non-chemical process and hence easily applicable and repeatable is significant. The authors also demonstrate chemical functionality by grafting a PEG-thiol brush and observing superior resistance against bacteria attachment. The convenience of the processing step and the enhanced properties of the modified PDA coating together represent a significant advancement, and this report should be considered for publication after addressing some minor comments listed below.

1) What is the relationship between the initial PDA coating thickness and the final thickness? The later descriptions around line 143 and Fig.3a imply that the 37 nm LAPDA layer came from something similar to the 248 nm PDA control. Would this always be the case? Can the authors clarify this thickness change when the annealing process is first described?

2) Assuming a thickness change is an integral part of the laser process, is there a minimum or maximum initial PDA thickness at which this LAPDA process may be applied? Also, it would be useful to discuss how the improvement in hardness and smoothness might change with the initial thickness of the PDA film.

3) Can the authors clarify how much the lack of scratch resistance for the regular PDA relates to the breaking free of the undesired PDA particle top "layer" and how much to the actual PDA base layer please? As the authors show in Fig.3a, the native PDA film is composed of a 100 nm base film plus particles extending a few hundreds of nanometers on top. Considering only the base film, which is probably more robust and more relevant to more common thinner PDA layers, how prone to scratching is the untreated smooth PDA?

4) In comparing the scratch resistance with TiO₂ and quartz, can the authors discuss how the tribological properties of the surfaces might affect the scratch test results, which focus on the scratch depth. How much do the differences shown have to do with the friction against the diamond tip vs. the hardness of the surfaces, as implied by the authors? Both hardness and low friction are actually quite beneficial properties.

5) Can the authors present further results demonstrating the chemical functionality of the LAPDA layer?

6) Fig.4b shows relatively poor antifouling for the PDA control. However, what is the level of E coli attachment on PEG attached on a thinner PDA layer that is as smooth as the LAPDA? This does not detract from the current achievement on the LAPDA, but rather bolster the argument for the improvement possible with smoother LAPDA layers.

Minor comments:

5) Please specify how molecular mass is measured in the caption for Fig.1f and g (state technique--MALDI MS)

6) Please revise labeling of Fig.2e to follow left to right, up to down convention.

7) Can the description surrounding Figure 1a in the main text briefly give the initial deposition conditions and make clear the initial thickness of the relevant PDA film.

8) It may be more more convenient for readers to translate to their own system if slightly different ways of presenting the laser annealing parameters were given. Is the stated 50 mm/s raster speed single line speed or the speed that the laser was scanned down the 1" wide quartz sample? How many passes were needed to accumulate the 280 microsecond dwell time? What was the diode laser power rating? Was this a continuous wave laser or a pulsed laser? Can the authors give the illumination dose in terms of energy, e.g. mJ/cm² (power/area x time), as is common with UV curing of adhesives?

9) It is mentioned that antifouling was enabled over large wafer sized areas. If larger wafers were prepared (larger than the 1x1" quartz sample), can the authors specify in the materials section?

Reviewer #2 (Remarks to the Author):

This work reports an interesting update on the processing of polydopamine (PDA) coating, where the laser annealed PDA exhibited improved scratch resistance and reduced surface roughness. The reviewer has the following comments

1) In the introduction, the authors stated that the ease of delamination of PDA from substrates could be attributed to the weak intermolecular bonds between PDA building blocks. Please elaborate why weak bonding between PDA units will lead to weak bonding between PDA and substrates.

2) The authors showed the simulation results of temperature profiles (Fig. S5). Does this result reflect the interfacial thermal resistance? Can you provide an estimate of the energy density absorbed by the PDA layer under different power levels? Was oxidation a concern in this process?

3) Figure 4 implies the existence of cracks in PDA films. If so, how wide were these cracks? Could these cracks be healed by the laser annealing process?

4)Wear resistance was reported in terms of penetration depth during the scratch test. It will be important to include the original load curves, which would show the vertical and horizontal forces as

well as the displacement as a function of loading time. Also, it is important to report the hardness values, which are closely related to the wear resistant property of coatings. Quartz and TiO₂ are hard materials and possess good wear resistance. It will be interesting to compare the hardness of the LAPDA with these materials.

5) Good wear resistance does not guarantee good adhesion to substrates, which as stated by the authors is a weakness of PDA coating. Can the authors measure the adhesion between the LAPDA layer and the substrate and make comparison to as-grown PDA?

6) Since furnace annealing can also induce partial graphitization, will it lead to similar improvement in wear resistance? Can you perform the same experiments on furnace annealed PDA films?

Reviewer #3 (Remarks to the Author):

This paper presents an interesting method for mechanically strengthened and smoothed polydopamine film preparation through the blue diode laser annealing process which induces the partial graphitization for the mechanically strong laser annealed polydopamine film formation. This study provides a unique and novel research on the polydopamine modification at the level that has never been achieved so far. I strongly support the publication of this paper after the following suggestions are reflected in the revision

1) For the laser process, specific laser parameter selection plays a very critical role due to the monochromatic and local nature of the laser process. If the authors can provide a very brief discussion on the reason why the specific laser parameters (elliptic laser profile, beam diameter, wavelength, power, etc) were chose, it will be very helpful to the other researchers.

2) Continuing previous comments, most laser process use circular beam shape. I am curious if there is any specific reason why an elliptic laser profile was used for this study because it is not usual laser configuration. Is it to simulate the line beam shape to cover the larger area?

3) I wonder what is the mechanism of the LAPDA process. I guess the authors want to say photo-thermal effect is the main mechanism because they did the numerical simulation. Is this by photo-thermal or photo-chemical? Blue wavelength is usually used for the photo-chemical process.

4) 100 fold scratch resistance increase after laser ablation looks very impressive. If the authors can briefly compare with other conventional methods to increase scratch resistance, it will be more useful to the readers.

5) The numerical simulation for the temperature rises during the laser irradiation on the PDA film does not provide quantitative information for the real temperature. I know it is extremely difficult to directly measure the temperature rise but some reference studies and other indirect measurement will be helpful to justify the temperature rise.

6) For the laser condition, instead of laser power, laser intensity information will be more useful.

Authors' Response to Reviewers' Comments

The authors are grateful to the reviewers for their interest in the paper and for their constructive comments and suggestions, which resulted in significant improvement of the revised manuscript. To facilitate checking the revisions made, a detailed point-by-point response is given below, along with the location in the revised manuscript where the corresponding revision was made. In addition, main revisions have been highlighted in the revised text of the manuscript. (Review comments are shown in black fonts, author responses are shown in blue font, and modifications are indicated in purple font and yellow highlight).

Reviewer #1 (Remarks to the Author):

Lee and co-workers report a novel processing method for coating surfaces with an ultra scratch resistant and chemically functional layer of polydopamine. The technique relies on diode laser induced annealing of a pre-coated layer of polydopamine (PDA). Superior scratch resistance is a result of graphitization and smoothing of the PDA layer. The simplicity of adjusting laser power to control graphitization and hence-wear resistance is very beneficial. The fact that this is an ambient, non-chemical process and hence easily applicable and repeatable is significant. The authors also demonstrate chemical functionality by grafting a PEG-thiol brush and observing superior resistance against bacteria attachment. The convenience of the processing step and the enhanced properties of the modified PDA coating together represent a significant advancement, and this report should be considered for publication after addressing some minor comments listed below.

1) What is the relationship between the initial PDA coating thickness and the final thickness? The later descriptions around line 143 and Fig.3a imply that the 37 nm LAPDA layer came from something similar to the 248 nm PDA control. Would this always be the case? Can the authors clarify this thickness change when the annealing process is first described?

We agree that the thickness change from the initial PDA thickness to the final LAPDA thickness was not clearly stated in the manuscript. Hence, in accordance with your comment, we further modified the manuscript as follows.

Page 9 and Line 19 in Manuscript:

“The pristine PDA film shows an average thickness of 248 nm with high roughness (~230 nm) due to the agglomerated NPs (Figure 3a). In contrast, after the laser annealing under 1.9 W, most NPs on the surface were ablated, resulting in exceptionally smoothed surface with 16 nm roughness and decreased thickness with 37 nm (Figure 3b), which is also identified in macroscopic and microscopic scales (Figures 3d and S9).”

2) Assuming a thickness change is an integral part of the laser process, is there a minimum or maximum initial PDA thickness at which this LAPDA process may be applied? Also, it would be useful to discuss how the improvement in hardness and smoothness might change with the initial thickness of the PDA film.

Thank you for your comment. The lower and upper thickness bounds can vary depending on the experimental parameters, thus our answers below relate to current experimental conditions, i.e. spot size of irradiated laser ($14\ \mu\text{m} \times 99\ \mu\text{m}$), applied laser power (1.9 W), and the scanning speed (50 mm/s equivalent to 280 μs dwell time).

Regarding the *maximum* initial PDA thickness, it can be estimated by a thermal diffusion length ($\sim\sqrt{at}$) that absorbed heat can penetrate along the thickness direction and induce the graphitization process. Here, t is dwell time (= 280 μs)

and α is the thermal diffusivity dictated by $\alpha = \frac{k}{\rho C}$, where k is the thermal conductivity, ρ is the density, and C is the heat capacity. By utilizing thermophysical properties of melanin (thermal conductivity $\sim 0.63 \text{ W m}^{-1} \text{ K}^{-1}$, density $\sim 2000 \text{ kg m}^{-3}$ and heat capacity $\sim 2500 \text{ J kg}^{-1} \text{ K}^{-1}$), the thermal diffusivity of PDA film is $1.26 \times 10^{-7} \text{ m}^2 \text{ s}^{-1}$ and the corresponding thermal diffusion length is expected to be $\sim 6.0 \text{ }\mu\text{m}$, which is larger than the optical penetration depth ($\sim 1.8 \text{ }\mu\text{m}$). Therefore, we may expect that $\sim 6.0 \text{ }\mu\text{m}$ in thickness would be the maximum for our current experimental conditions, and that PDA films thicker than $\sim 6.0 \text{ }\mu\text{m}$ would have the remaining layer unaffected by the absorbed heat. (i.e., the entire film will not have a uniform partially-graphitized film). However, lower scanning speed ($< 50 \text{ mm/s}$) will allow the larger maximum thickness due to the longer thermal diffusion length, as predicted in the same manner.

With respect to the *minimum* thickness, this could be limited by the absorbance of PDA films. As discussed in Figure S5 and Figure S10, the absorbance of PDA film with 100 nm thickness is 0.05. Since the absorbance exponentially decreases as the film thickness decreases, sufficient thickness of PDA film is required to drive the temperature high, which will be known as the “minimum thickness”. For example, when we consider 1.2 W annealing processes that started inducing the partial graphitization shown in Figure 1, the absorbed laser energy is $0.05 \times 1.2 \text{ W} = 0.06 \text{ W}$, and the annealing temperature is estimated to be 1000 K. If we use 1.9 W laser power and decrease the thickness until the absorbed energy reaches 0.06 W, then the relationship can be expressed as follows.

$$0.06 = 1.9 \times \int_0^{\delta} \alpha e^{-\alpha z} dz$$

where α is the absorbance, and δ is the thickness of PDA films. By solving the above equation, the minimum thickness of PDA films would be $\sim 56 \text{ nm}$ for 1.9 W annealing process. Again, the value of minimum thickness will be significantly affected by the imparted laser energy and the absorbance (thickness).

Lastly, regarding the impact of initial thicknesses on improvement in the smoothness, the main contributor for roughness is the randomly agglomerated NPs, as discussed in Figure 3. Hence, the authors believe that the initial thickness of PDA films would not have a huge impact on the improvement in the smoothness.

Page 9 and Line 19 in Manuscript:

“The pristine PDA film shows an average thickness of 248 nm with high roughness ($\sim 230 \text{ nm}$) due to the agglomerated NPs (Figure 3a). In contrast, after the laser annealing under 1.9 W, most NPs on the surface were ablated, resulting in exceptionally smoothed surface with 16 nm roughness and decreased thickness with 37 nm (Figure 3b), which is also identified in macroscopic and microscopic scales (Figures 3d and S9). We expect that the initial thickness of the PDA film would not have a huge impact on the improvement in the smoothness since the main contributor for roughness is the randomly agglomerated NPs.”

With respect to the hardness, please refer to the authors’ response to comment #4 from Reviewer #2 (see below).

3) Can the authors clarify how much the lack of scratch resistance for the regular PDA relates to the breaking free of the undesired PDA particle top "layer" and how much to the actual PDA base layer please? As the authors show in Fig.3a, the native PDA film is composed of a 100 nm base film plus particles extending a few hundreds of nanometers on top. Considering only the base film, which is probably more robust and more relevant to more common thinner PDA layers, how prone to scratching is the untreated smooth PDA?

Thank you very much for this comment, and we completely agree that the existence of undesired PDA particles may affect the overall scratch resistance. Unfortunately, we know of no method for producing untreated smooth PDA of this thickness, as roughness is inherent to all thick PDA films using our method. However, we can infer that if the particles played a particular role in scratch resistance along with the base layer, a more substantial fluctuation in the scratch curves would be expected due to the presence of PDA particles. However, as displayed in Figure 2a, scratch depths for PDA films, during the loading time show a non-fluctuating profile ($18.5 \pm 1.1 \text{ nm}$), suggesting that the

indenter is either passing through the PDA particles or is dragging the particles into the bulk of the coatings. Besides, there is no direct evidence that the PDA particles present on the top of the coating exhibit different mechanical properties compared to the bulk PDA. We would like to emphasize that the data should be viewed in comparison (pristine PDA vs. LAPDA); indeed, our scratch experiments reveal that 50 μN of load cannot result in any scratch damage in LAPDA, in stark contrast to what we observe in the pristine PDA.

4) In comparing the scratch resistance with TiO_2 and quartz, can the authors discuss how the tribological properties of the surfaces might affect the scratch test results, which focus on the scratch depth. How much do the differences shown have to do with the friction against the diamond tip vs. the hardness of the surfaces, as implied by the authors? Both hardness and low friction are actually quite beneficial properties.

Thank you very much for the thoughtful comment. Indeed, the tribological properties depend on the elastic-plastic material properties, such as the elastic modulus and hardness, and the surface topography and adhesion characteristics. The mechanical properties control the scratch depth and dominant damage (wear) mechanism (e.g., adhesion, abrasion, delamination, surface cracking) of the scratched PDA coating. High friction traction generates high tensile stresses that are conducive to surface and subsurface cracking (cohesive failure) of the coating. During laser annealing, the topmost layer containing the particles is removed and the base layer of the PDA undergoes structural and morphological changes, partially converting it to a graphitic material exhibiting a higher degree of covalent crosslinking and crystallinity than pristine PDA. The enhanced crystallinity increases the overall strength and stiffness of the coating. In addition, considering the friction force comprises contributions from interface shearing (adhesion friction component) and bulk shearing (plowing friction component), LAPDA should exhibit higher scratch resistance compared to pristine PDA and possibly lower friction due to the lack of the particle top layer that induces roughening. On the other hand, as shown in Figure 2d, after the indenter was fully retracted, a scratch was observed on the surface, suggesting permanent deformation, implying that hardness (defined at fully plastic deformation conditions) also contributes to the overall behavior. A rough estimation of the hardness of LAPDA with respect to TiO_2 and quartz shows that the hardness of LAPDA is similar or exceeds the hardness of TiO_2 and quartz; however, defining a single hardness value for PDA or LAPDA is probably impossible due to the heterogeneous nature of this material. Nevertheless, we believe that a combination of low friction and increased hardness should combine to produce the scratch resistance observed.

5) Can the authors present further results demonstrating the chemical functionality of the LAPDA layer?

Thank you for your comment. To further demonstrate the remaining catechol functionality, we performed electroless metallization on the LAPDA layer. Briefly, catechol has an ability to reduce metallic ions, resulting in a metallic layer on the catechol-functionalized surface (Lee et al., Science 2007, 318, 426). For this, the LAPDA film was dipped in silver nitrate solution (50 mM) for 18 hours with the mild shaking. Bare quartz was also treated with the silver nitrate solution as a control sample. Based on XPS analysis shown in Figure S13, the metallic film (silver) was successfully deposited only on the LAPDA substrate, which confirms that the inherent catechol functionality is preserved on the LAPDA surface.

Figure S13. Demonstration of catechol functionality through electroless metallization on LAPDA. The schematic of 50 mM silver nitrate solution treatment on a) bare quartz and b) LAPDA on quartz; and their corresponding X-ray photoelectron spectroscopy analyses were described on the bottom of each scheme. The high-resolution Ag3d data was additionally collected from the treated LAPDA substrate, and presented at the left top of the corresponding plot.

The experimental details and related discussions are newly added to the revised manuscript as follows.

Page 12 and Line 2 in Manuscript:

“Moreover, an additional example of secondary surface modification via electroless metallization on LAPDA was presented in Figure S13.”

6) Fig.4b shows relatively poor antifouling for the PDA control. However, what is the level of E coli attachment on PEG attached on a thinner PDA layer that is as smooth as the LAPDA? This does not detract from the current achievement on the LAPDA, but rather bolster the argument for the improvement possible with smoother LAPDA layers.

Thanks for your comment. In addition to 248 nm average thickness of PDA film (Figure 3a), a thinner PDA layer that is as smooth as the LAPDA was prepared by reducing the coating time (Figure S14a). The PDA-coated substrate was subsequently PEGylated by following the protocols described in the Methods section. It was found that the physical delamination occurred in the thin film (Figure S14b) as comparable as in the thick film (Figure 4b), which similarly resulted in the partial failure of antifouling from the bacterial adhesion. Given that the PEGylated LAPDA film sharing similar thickness and smoothness of the thin PDA film showed even better antifouling performance (Figure 4a) compared to the thick PEGylated PDA film (Figure 4b), the above result once again highlights the positive effect of mechanically strengthened LAPDA film. As requested by the reviewer, we added the above-mentioned experimental result to revised the manuscript.

Figure S14. (a) AFM image with line-scan profile of pristine PDA deposited for 2 hours, showing 25 nm average thickness (20 nm flat PDA film with 20 nm roughness). (b) The bright-field (upper) and fluorescence (lower) images of bacterial attachment on the PEGylated PDA film after 24 hours of incubation. The blue arrows indicate bacterial accumulation on sites of mechanical damage in the PEGylated PDA coating. The scale bar is 200 μm .

Page 12 and Line 8 in Manuscript:

“As shown in **Figure 4a**, the bare quartz substrate (positive control) was fully occupied by *E. coli* (green spots) within 24 hours. In contrast to pristine PDA which performed poorly due to weak mechanical adhesion to the substrate (Figures 4b and S14), under the same conditions the PEGylated LAPDA film remained smooth, mechanically stable and showed successful inhibition of bacterial attachment (94% reduction relative to bare quartz) (Figures 4c and S15).”

Minor comments:

5) Please specify how molecular mass is measured in the caption for Fig.1f and g (state technique--MALDI MS)

Thank you for your comment, and the authors described the requested contents in the manuscript.

Page 4 and Line 6 in Manuscript; Caption of Figures 1f and g:

“MALDI-TOF molecular mass analysis of (f) PDA film and (g) LAPDA film at 1.9 W.”

6) Please revise labeling of Fig.2e to follow left to right, up to down convention.

Thank you for your comment. We revised the labeling of Figure 2e accordingly.

7) Can the description surrounding Figure 1a in the main text briefly give the initial deposition conditions and make clear the initial thickness of the relevant PDA film.

Thank you for your comment. We revised the description accordingly.

Page 3 and Line 19 in Manuscript:

“Shown in Figure 1a, Figure S1, and Video S1 are schematics of the **continuous wave** blue diode laser annealing process for the PDA film in the ambient air condition, **and PDA film with 248 nm average thickness was prepared by dip-coating quartz substrate in the aqueous dopamine hydrochloride solution for 24 hours (see details in the Methods section)**. The focused blue diode laser has an elliptically shaped beam profile with 99 μm long axis and 14 μm short axis based on e² beam diameter, measured by a knife edge experiment.”

8) It may be more more convenient for readers to translate to their own system if slightly different ways of presenting the laser annealing parameters were given. Is the stated 50 mm/s raster speed single line speed or the speed that the laser was scanned down the 1” wide quartz sample? How many passes were needed to accumulate the 280 microsecond dwell time? What was the diode laser power rating? Was this a continuous wave laser or a pulsed laser? Can the authors give the illumination dose in terms of energy, e.g. mJ/cm² (power/area x time), as is common with UV curing of adhesives?

The authors appreciate your comment. The dwell time is defined by (= the length of short axis / scanning speed). In our experiments, a single pass was utilized to achieve 280 μs dwell time [= 14 (μm) / 50 (mm/s)], and we utilized the continuous wave blue diode laser. To improve clarity, we modified the main text, illustrated the aforementioned annealing process in Figure S1b, and added the laser conditions in intensity (kW/cm²) in the Supporting Information.

Page 3 and Line 19 in Manuscript:

“Shown in **Figure 1a**, Figure S1, and Video S1 are schematics of the **continuous wave** blue diode laser annealing process for the PDA film in the ambient air condition, **and PDA film with 248 nm average thickness was prepared by dip-coating quartz substrate in the aqueous dopamine hydrochloride solution for 24 hours (see details in the Methods section)**. The focused blue diode laser has an elliptically shaped beam profile with 99 μm long axis and 14 μm short axis based on e² beam diameter, measured by a knife edge experiment. Annealing of a PDA film deposited on a 1” by 1” quartz substrate was accomplished by raster-scanning with a speed of 50 mm/s (equivalent to dwell time of 280 μs = $\frac{\text{short axis}}{\text{scanning speed}}$). The scan lateral separation was set at 35 μm in order to achieve uniform annealing on the overlapped region, as illustrated in Figure S1b.”

Page 1 and Line 14 in Supporting Information:

Figure S1. The schematics of (a) the optical setup for the BLA process, and (b) the raster scanning method to anneal the PDA film.

Explanation for Figure S1

We utilized the fundamental (elliptical) laser source to demonstrate the capability of large area processing via a line beam shape. Moreover, an elliptically shaped beam profile with 99 μm long axis and 14 μm short axis is employed to obtain sufficient laser intensities (kW/cm^2) within the output of a single diode laser in order to ensure the temperature rise.

Laser Power (W)	Laser Intensity (kW/cm^2)
0.9	20.7
1.2	27.6
1.9	43.6
2.4	55.1
2.9	66.6

Table S1. Laser conditions in intensity (kW/cm^2)

9) It is mentioned that antifouling was enabled over large wafer sized areas. If larger wafers were prepared (larger than the 1x1" quartz sample), can the authors specify in the materials section?

Thank you for your comment. Although the largest size of quartz substrate we prepared was a 1" by 1" herein, we truly believe that PDA films larger than 1" by 1" can be easily converted to LAPDA films due to scalable and facile processing protocols rendered by the raster scanning method, as displayed in Figure S1b. Furthermore, for high throughput processing multiple diode lasers can be coupled to the apparatus, thereby increasing the irradiated power, extending the lateral width of the beam (comment in the summary). However, the authors fully acknowledge that 'wafer-sized area' may be misleading to readers, so we specified the size we tested in order to make the statement clearer.

Page 3 and Line 14 in Manuscript:

"Unlike PEGylated PDA films that were prone to scratch defects, the PEGylated LAPDA preserved antifouling characteristics over an entire wafer-sized area (1" by 1")."

Page 12 and Line 22 in Manuscript:

"From the processing point of view, the film annealing throughput can be increased since multiple blue diode laser modules can be combined to form a wide beam line of high power."

Reviewer #2 (Remarks to the Author):

This work reports an interesting update on the processing of polydopamine (PDA) coating, where the laser annealed PDA exhibited improved scratch resistance and reduced surface roughness. The reviewer has the following comments

1) In the introduction, the authors stated that the ease of delamination of PDA from substrates could be attributed to the weak intermolecular bonds between PDA building blocks. Please elaborate why weak bonding between PDA units will lead to weak bonding between PDA and substrates.

Thank you for your comment. The authors agree that the term ‘delamination’ may be misleading in this case since it could be interpreted as an interfacial separation coming from the weak bonding between PDA film and underlying substrate. To further clarify the statement, we revised the manuscript as follows:

Page 2 and Line 5 in Manuscript:

“The first is inherently poor mechanical properties of PDA, which inevitably results in **continuous abrasion** of PDA film, **exposing** the underlying substrate **in the end**². Considering the current understanding of the formation and molecular structure of PDA^{10, 11, 12}, this deficiency could be attributed to the relatively weak non-covalent intermolecular bonds between PDA building blocks^{13, 14, 15}.”

2) The authors showed the simulation results of temperature profiles (Fig. S5). Does this result reflect the interfacial thermal resistance? Can you provide an estimate of the energy density absorbed by the PDA layer under different power levels? Was oxidation a concern in this process?

The authors greatly appreciate the reviewer’s comment. The simulation results in Figure S5 displays the numerically simulated temperature profiles of only the flat PDA film under different laser powers, and the interfacial thermal resistance we referred to in the manuscript indicates the thermal contact resistance between the flat PDA film and PDA NPs. Since it is difficult to extract the aforementioned contact resistance due to random distributions, agglomerations, and high number density of NPs, we did **not** include the resistance of PDA NPs in 3D numerical simulations. Instead, we performed the independent thermal analysis via 3D numerical simulations for the flat PDA film (Figure S5), and Mie theory / lumped capacitance analysis (Figures S10 and S11) for PDA NPs.

The estimated energy deposited on the flat PDA layer is represented by “absorptance”, which is a normalized ratio of absorbed energy over the incident laser energy, and this dimensionless number is estimated to be 0.05 for the 100 nm PDA film. Hence, absorbed laser energy on PDA films are 0.05 W, 0.06 W, 0.1 W, 0.12 W, and 0.15 W, under the incident laser power 0.9 W, 1.2 W, 1.9 W, 2.4 W, and 2.9 W, respectively.

Based on your comment, we provided estimated energy absorbed by the PDA layer in the Supporting Information.

Page 6 and Line 22 in Supporting Information:

“The numerical simulation performed accounts only for the laser absorption and heat diffusion to characterize the upper bound of the transient temperature of the PDA film **under different laser powers (0.9 W, 1.2 W, 1.9 W, 2.4 W, and 2.9 W)**. The corresponding absorbed laser powers on PDA films are 0.05 W, 0.06 W, 0.1 W, 0.12 W, and 0.15 W, **respectively**. The modeling of the phase change mechanism from amorphous to poly-crystalline graphitization was not included herein.”

3) Figure 4 implies the existence of cracks in PDA films. If so, how wide were these cracks? Could these cracks be healed by the laser annealing process?

Thank you for your comment. We would like to emphasize that images shown in Figure 4 correspond to the appearance of surfaces subjected to multiple sample treatment steps including pipetting, vibration, and shaking. Cracking and loss of PDA films are mainly attributed to spontaneous loss of portions of the PDA films during the experiment.

Nevertheless, to answer your question whether scratches in pristine PDA films might be healed by BLA, we intentionally made a scratch of $\sim 1.5 \mu\text{m}$ width as shown in Figure R1a, performed BLA across the scratch area, and then analyzed the region again by AFM. The result showed that for the scratch width tested ($\sim 1.5 \mu\text{m}$), healing of the scratch by laser annealing did not occur (Figure R1b).

Figure R1. Fully penetrated scratch tracks by AFM for (a) the pristine PDA film and (b) the LAPDA film. The width of scratch was $\sim 1.5 \mu\text{m}$ before BLA process, and was largely unchanged after BLA treatment ($\sim 1.5 \mu\text{m}$).

4) Wear resistance was reported in terms of penetration depth during the scratch test. It will be important to include the original load curves, which would show the vertical and horizontal forces as well as the displacement as a function of loading time. Also, it is important to report the hardness values, which are closely related to the wear resistant property of coatings. Quartz and TiO_2 are hard materials and possess good wear resistance. It will be interesting to compare the hardness of the LAPDA with these materials.

Thank you very much for the thoughtful feedback and suggestions. Representative raw data illustrating the lateral load, normal load and normal displacement as a function of loading time for pristine PDA at $50 \mu\text{N}$ and LAPDA at $500 \mu\text{N}$ are shown in Figure S7.

Page 6 and Line 25 in Manuscript:

“Figures 2a and S7a summarize the scratching experiments using $50 \mu\text{N}$ normal load performed on the pristine PDA film and the LAPDA film annealed at 1.9 W. The average scratch depth for the pristine PDA at $50 \mu\text{N}$ normal load was $18.5 \pm 1.1 \text{ nm}$, whereas the LAPDA films were undamaged at this load. Scratches in LAPDA films were only observed at much higher load as shown in Figures 2b and S7b.”

Page 9 and Line 14 in Supporting Information:

Figure S7. Representative raw data illustrating the lateral load, normal load and normal displacement as a function of loading time for (a) pristine PDA at 50 μN and (b) LAPDA at 500 μN .

We can estimate the hardness of LAPDA in comparison to TiO_2 and Quartz by evaluating the scratch measurements that we performed. According to ASTM G171 (ASTM G171 standard test method for scratch hardness of materials using a diamond stylus, ASTM Stand. 2017, 3), this hardness test utilizes the residual scratch width after the indenter is removed reflecting the permanent deformation resulting from scratching and not the instantaneous state of combined elastic and plastic deformation of the surface. The scratch hardness number can be calculated by dividing the applied normal load by the projected area of scratching contact by using the following equation:

$$H = 8P / \pi w^2$$

where P = normal force, and w = scratch width. Representative images of the scratch tracks after the test was performed are shown in Figure 2. TiO_2 exhibits the highest width (500 nm in this case) followed by quartz and finally LAPDA with 450 nm and 380 nm respectively, resulting in calculated hardness values of $H_{\text{TiO}_2} = 4.6 \pm 0.4$ GPa, $H_{\text{Quartz}} = 6.2 \pm 0.9$ GPa, and $H_{\text{LAPDA}} = 6.6 \pm 1.0$ GPa. Among the substrates, LAPDA has the highest hardness, which is evident in the relevant profile images. The above values were taken as average values out of four measured widths for each sample.

Figure S8. (a) Scratch tracks for TiO₂, quartz and the LAPDA after the scratch measurements, and (b) their corresponding images converted by Gwyddion software for hardness calculation. The scale bar is 2 μm.

Page 7 and Line 7 in Manuscript:

“To further put the scratch resistance of LAPDA in perspective, we also performed scratching experiments on bare TiO₂ and quartz substrates at a load of 500 μN, producing average scratch depths of 16.8 ± 1.2 nm for TiO₂ and 10.5 ± 1.0 nm for quartz. Surprisingly, this indicates that the exceptional scratch resistance of the organic LAPDA film is several-fold better than even hard inorganic solids. Moreover, the calculated hardness of LAPDA (6.6 ± 1.0 GPa) was superior to that of TiO₂ and quartz, showing 4.6 ± 0.4 GPa and 6.2 ± 0.9 GPa, respectively (Figure S8).”

5) Good wear resistance does not guarantee good adhesion to substrates, which as stated by the authors is a weakness of PDA coating. Can the authors measure the adhesion between the LAPDA layer and the substrate and make comparison to as-grown PDA?

Indeed, adhesion is a multivariate property that depends on many parameters such as surface energy, chemical, and mechanical properties of the coating. However, for pristine PDA, which exhibits a high degree of heterogeneity and roughness, such a study would be complex and beyond the scope of this work. Nevertheless, in one of our previous studies where we performed moderate thermal annealing of PDA at 400 K we investigated the molecular mechanics of pristine and annealed PDA via single-molecule force spectroscopy, and we showed that thermal annealing enhanced the intermolecular and cohesive interactions of the film (ACS Appl. Mater. Interfaces 2019, 11, 46, 43599). Here, a similar trend is observed in the LAPDA film, where the laser annealing enhances crosslinking and crystallinity of the film resulting in a more robust material with better shear resistance. Overall, our study reveals that the coating cohesion is improved; however, we cannot thoroughly comment on the adhesion. While it would be interesting to evaluate the adhesion of the PDA with the substrates prior to and after laser annealing, we believe that it will be outside of the scope of the current study.

6) Since furnace annealing can also induce partial graphitization, will it lead to similar improvement in wear resistance? Can you perform the same experiments on furnace annealed PDA films?

Thank you for your insightful comments. In our previous study, we demonstrated that the PDA annealed at ~ 400 K furnace resulted in improvement in wear-resistance (ACS Appl. Mater. Interfaces 2019, 11, 46, 43599). However, the improvement in wear resistance was far less effective (~2.5 fold increase compared to pristine PDA) compared to the current result (~100 fold increase). The post-treatment of PDA in high annealing temperature may further enhance its wear resistance, however, there is a significant drawback for this approach. According to previous studies (Angew. Chem. 2013, 125, 5645–5648; ACS Appl. Mater. Interfaces 2017, 9, 6655–6660), PDA thermally annealed at 800 to 1000 °C resulted in ‘fully’ graphitized PDA surface as measured by XPS and EDS, which makes the material less attractive due to the loss of important PDA-like properties. Thus, we feel that our approach is more attractive.

Reviewer #3 (Remarks to the Author):

This paper presents an interesting method for mechanically strengthened and smoothed polydopamine film preparation through the blue diode laser annealing process which induces the partial graphitization for the mechanically strong laser annealed polydopamine film formation. This study provides a unique and novel research on the polydopamine modification at the level that has never been achieved so far. I strongly support the publication of this paper after the following suggestions are reflected in the revision

1) For the laser process, specific laser parameter selection plays a very critical role due to the monochromatic and local nature of the laser process. If the authors can provide a very brief discussion on the reason why the specific laser parameters (elliptic laser profile, beam diameter, wavelength, power, etc) were chose, it will be very helpful to the other researchers.

The authors greatly appreciate the reviewer’s comment, and we further modified the manuscript according to your comment to make it helpful to other researchers.

Page 2 and Line 22 in Manuscript:

“In particular, blue diode laser²² has been introduced in thin film annealing owing to its scalability, continuous wave characteristics, high optical power density per unit area/volume, low cost and high photon energy (2.82 eV)^{23, 24}. Considering the light absorption of PDA in the range of blue-diode laser wavelength (440 nm)¹⁵, we inferred that blue-diode laser annealing (BLA) could be employed to efficiently anneal and improve PDA properties.”

With respect to the laser profiles and the diameter, please refer to the Comment #2.

Regarding the selection of laser powers, various laser powers were employed to investigate their effect on the graphitization processes.

Page 4 and Line 9 in Manuscript:

“To corroborate the partial graphitization by the BLA process, the Raman spectra of the pristine PDA and the LAPDA films with various laser-annealing powers (0.9, 1.2, 1.9, 2.4, and 2.9 W) were compared (Figure 1b), and the corresponding laser condition in intensity (kW/cm²) is presented in Table S1.”

2) Continuing previous comments, most laser process use circular beam shape. I am curious if there is any specific reason why an elliptic laser profile was used for this study because it is not usual laser configuration. Is it to simulate the Line beam shape to cover the larger area?

Thank you for your comment, and there are two main reasons why we used an elliptical laser beam profile. First of all, a blue diode laser is a semiconductor diode laser which has an asymmetrical rectangular slit where the photons are emitted from its body. Due to this geometric configuration, the fundamental laser beam profile has an elliptical cross section in the far field which is different from the other laser sources with a circular Gaussian laser beam profile. Second, as you mentioned, we utilized this fundamental (elliptical) laser source as it is to show the capability of a large area processing via a Line beam shape. Moreover, an elliptically shaped beam profile with 99 μm long axis and 14 μm short axis is employed to obtain sufficient laser intensities (kW/cm^2) within the output of a single diode laser in order to ensure the temperature rise (i.e., partial graphitization). Furthermore, given the major axis length is substantially larger than the pitch between scans to ensure uniform processing over the film area. Based on your comment, we further specified the reasons why the elliptical beam profile and the beam diameter were utilized.

Page 1 and Line 19 in Supporting Information:

Explanation for Figure S1

We utilized the fundamental (elliptical) laser source to demonstrate the capability of large area processing via a line beam shape. Moreover, an elliptically shaped beam profile with 99 μm long axis and 14 μm short axis is employed to obtain sufficient laser intensities (kW/cm^2) within the output of a single diode laser in order to ensure the temperature rise.

3) I wonder what is the mechanism of the LAPDA process. I guess the authors want to say photo-thermal effect is the main mechanism because they did the numerical simulation. Is this by photo-thermal or photo-chemical? Blue wavelength is usually used for the photo-chemical process.

The authors sincerely appreciate the reviewer's comment. As mentioned by the reviewer, there are two mechanisms in the range of near UV that may affect the graphitization. The contributions of these mechanisms depend upon the imparted wavelength, pulse duration (or dwell time), and materials. Hence, we screened the LAPDA process to verify which process leads the graphitization for the blue wavelength (440 nm) by varying the scanning speed. If the dominant mechanism was driven by the photochemical process, similar results would be obtained by reducing the laser power and increasing the exposure time (dwell time), as is the case in typical UV curing of polymers.

Thus, we performed new experiments under the same laser intensities but with lower scanning speed (18 mm/s equivalent to 778 μs dwell time). Based on Raman spectroscopy in Figure S6a, ~ 2.8 times more dwell time with 0.9 W did **not** play a critical role in the graphitization, but laser powers higher than 1.2 W started inducing partial graphitization. To further put the estimated temperature in perspective, numerical simulations at 0.9 W, 1.2 W, and 1.9 W with 18 mm/s were performed and presented in Figure S6b. As a result, the maximum temperature under 0.9 W was anticipated to be 922 K (below 1000 K), and to be above 1000 K at 1.2 W, and 1.9 W, which is consistent with 50 mm/s annealing processes described in the manuscript.

Therefore, when considering all experimental and numerical analyses, we believe that the main mechanism of the LAPDA process rendered by BLA is photothermal rather than photochemical, leading to the partial graphitization.

Page 6 and Line 14 in Manuscript:

“To estimate the temperature field reached during laser annealing, the 3D heat conduction equation was numerically solved (Figure S5a). Figure S5b shows the time-dependent temperature profiles and Figure S5c indicates the temperature field contours at the surface of the PDA film. Detailed analysis regarding the photothermal process is discussed in Figure S6.”

Figure S6. (a) Graphitization level of each LAPDA film with 18 mm/s scanning speed. (b) Time-dependent temperature profiles at 'O' under different laser powers. Schematic of numerical simulations is shown in Figure S5a.

4) 100 fold scratch resistance increase after laser ablation looks very impressive. If the authors can briefly compare with other conventional methods to increase scratch resistance, it will be more useful to the readers.

The authors sincerely appreciate the reviewer's comment and have revised the manuscript to make it more useful to the readers.

Page 7 and Line 3 in Manuscript:

“At 500 μN, the scratch depth of a 37 nm thick LAPDA film was only 2.1 ± 0.8 nm whereas the scratch depth of a pristine PDA film under the same conditions was the full thickness of the film (248 nm). This represents more than a 100-fold increase in scratch resistance, which is more effective than the conventional thermal annealing approach showing a 2.5-fold increase¹⁶”

5) The numerical simulation for the temperature rises during the laser irradiation on the PDA film does not provide quantitative information for the real temperature. I know it is extremely difficult to directly measure the temperature rise but some reference studies and other indirect measurement will be helpful to justify the temperature rise.

The authors greatly appreciate the reviewer's comment and fully agree that the numerical simulation may not provide quantitative information for the real temperature. Hence, according to your remark, we added more references to support the temperature rise.

Page 5 and Line 6 in Supporting Information:

Explanation for Figure S5

When a continuous wave laser is irradiated on absorbing media, the absorbed energy contributes to increasing the temperatures, which can be experimentally probed and modeled by a heat conduction equation.^{1,2,3} Hence, to estimate the transient temperature field of the flat PDA film, a three-dimensional heat conduction equation was numerically solved by the finite difference method⁴. Because the flat PDA film is semi-transparent at 440 nm wavelength and its thickness (approximately 100 nm) is much thinner compared to the thermal diffusion length (approximately 20 μm at dwelling time of 280 μs and scanning speed of 50 mm/s), the entire PDA film is volumetrically heated by the blue

diode laser. Furthermore, most of the heat is dissipated into the substrate as the thermal penetration depth is much bigger than the film thickness.

6) For the laser condition, instead of laser power, laser intensity information will be more useful.

Thank you for your comment. Accordingly, the laser conditions power (W) as well as intensity (kW/cm²) were added to the Supporting Information.

Page 4 and Line 9 in Manuscript:

“To corroborate the partial graphitization by the BLA process, the Raman spectra of the pristine PDA and the LAPDA films with various laser-annealing powers (0.9, 1.2, 1.9, 2.4, and 2.9 W) were compared (Figure 1b), and the corresponding laser condition in intensity (kW/cm²) is presented in Table S1.”

REVIEWERS' COMMENTS:

Reviewer #1 (Remarks to the Author):

The authors have adequately addressed earlier comments. The manuscript should be considered for publication.

Reviewer #2 (Remarks to the Author):

The authors have addressed much of my concerns. I hope they can make one more improvement by commenting the following observation: the hardness of annealed PDA is higher than the ceramics included in their study. Is that possible that the LAPDA (at least partially) possesses diamond domains?

Reviewer #3 (Remarks to the Author):

The authors responded well to the comments. This should be ready for publication.

Authors' Response to Reviewer's Comment

Reviewer #2 (Remarks to the Author):

Question) The authors have addressed much of my concerns. I hope they can make one more improvement by commenting the following observation: the hardness of annealed PDA is higher than the ceramics included in their study. Is that possible that the LAPDA (at least partially) possesses diamond domains?

Answer) Thank you very much for the comment. Our results indicate the LAPDA has been partially converted into graphitic material, exhibiting a higher degree of covalent cross-linking and crystallinity than pristine PDA. While we agree that formation of diamond domains would be a provocative finding, we consider this too speculative without embarking on further studies (e.g. EELS, XRD) that are beyond the scope of this work.